# Lactobacillus Kefir M20 Adaptation to Bile Salts: A Novel Pathway for Cholesterol Reduction

**DOI:** 10.3390/foods13213380

**Published:** 2024-10-24

**Authors:** Changlu Ma, Qichen Liu, Shuwen Zhang, Ailing Qu, Qing Liu, Jiaping Lv, Xiaoyang Pang

**Affiliations:** 1College of Food and Bio-Engineering, Beijing Vocational College of Agriculture, Beijing 102442, China; machanglu@126.com (C.M.); 71629@bvca.edu.cn (A.Q.); 81905@bvca.edu.cn (Q.L.); 2Institute of Food Science and Technology, Chinese Academy of Agricultural Science, Beijing 100193, China; liuqichen1999@163.com (Q.L.); zhangshuwen@caas.cn (S.Z.); lvjiapingcaas@126.com (J.L.)

**Keywords:** lactobacillus kefir, acid stress adaption, bile stress adaption, cholesterol lowering, hamster

## Abstract

(1) Background: This study investigated the impact of in vitro adaptations to acid and bile stress on the cholesterol-lowering activity of the probiotic Lactobacillus kefir M20. (2) Methods: Lactobacillus kefir M20 was extracted from fermented dairy products in Xinjiang, China, and isolated using MRS medium. The lactic acid bacteria were cultured for stress resistance to acid and bile salts and then gavaged into mice for animal experiments. (3) Results: The adaptation to bile stress treatment resulted in a notable enhancement of the cholesterol-lowering capacity of Lactobacillus kefir M20, with reductions of 16.5% and 33.1% in total and non-HDL cholesterol, respectively, compared to the untreated strain. Furthermore, the daily fecal total bile acid excretion was 9.2, 5.4 and 5.0 times higher in the M20-BSA group compared to the HC, M20 and M20-ASA groups, respectively. (4) Conclusions: This study suggests that targeted probiotics have the potential for application in the next generation of functional foods and probiotic formulations aimed at combating hypercholesterolemia.

## 1. Introduction

Cardiovascular diseases have become the leading cause of mortality worldwide, and elevated serum cholesterol has been demonstrated to be the principal cause [1]. Due to the various adverse effects of commonly used cholesterol-lowering drugs such as statins, functional foods including probiotics have received more attention for their ability to modulate lipid metabolism [2]. When present in sufficient amounts, probiotics confer health benefits to the host. Several studies have focused on the cholesterol-lowering effects of probiotics [3].

*Lactic acid bacteria*, especially *Lactobacillus kefir M20*, are among the most commonly used probiotic microorganisms [4,5]. Several Lactobacillus strains have exhibited significant hypocholesterolemic activity in both experimental animals [6,7] and humans [8,9]. However, the exact mechanisms of action are not fully understood. Several possible mechanisms for the hypocholesterolemic effects of probiotic strains have been proposed, including cholesterol assimilation by the strains, cholesterol binding to the cellular surface of the strains, cholesterol incorporation into the cellular membrane [10], short-chain fatty acid production in the large intestine by the strains [11] and bile salt deconjugation by the strains with bile salt hydrolase activity [12].

Among these hypotheses, the most widely recognized mechanism for the hypocholesterolemic activity of probiotic strains is attributed to the bile salt hydrolase activity of the probiotic strains [13,14,15]. Clinical trials have shown that fermented milk with bile salt hydrolase-active *L. reuteri* exhibits significant hypocholesterolemic activity in hypercholesterolemic patients [8]. Additionally, oral administration of immobilized bile salt hydrolase derived from *L. buchneri* ATCC4005 caused a significant reduction in serum total cholesterol concentration by 58% in high-cholesterol-fed rats [16,17].

To exert hypocholesterolemic activity, Lactobacillus strains must reach a sufficient number of viable bacteria in the intestinal tract, requiring that the strains have greater gastrointestinal transit tolerance ability [18]. An in vitro study demonstrated that acid and bile stress adaptation treatment significantly improved the gastrointestinal transit tolerance of *L. kefiranofaciens* [19]. However, there are few studies on the role of in vitro stress-adaptive therapy in enhancing the cholesterol-lowering activity of Lactobacillus strains. Based on our investigations, many of the bacteria isolated from *kefir* possess probiotic properties, including the ability to lower serum and plasma cholesterol, angiotensin converting enzyme inhibitory activity, proven cardiac function, immunomodulatory properties and the ability to ameliorate non-alcoholic fatty liver disease and obesity.

## 2. Materials and Methods

### 2.1. Origin and Maintenance Condition of Strain

L. kefir M20, used in the present study, was isolated from a koumiss sample, a traditional fermented dairy product consumed in Xinjiang, China. The isolate was identified through 16S rRNA gene sequencing followed by a sequence similarity search with BLAST and carbohydrate fermentation pattern assay using an API-50CHL kit from BioMérieux (Craponne, France). The strain was maintained by subculturing in MRS broth (Oxoid, Hampshire, UK) with 1% inocula, using a 16 h incubation period at 37 °C. The strains were passaged three times in MRS broth under the above culture conditions for subsequent experiments (Major instruments and equipment and major test reagents see Appendix A). 

### 2.2. Stress Adaptation Treatments

According to a previous study [2], the sublethal levels for acid and bile stress of the strain were pH 4.5 and 0.05%, respectively. In vitro stress adaption conditions for the strain were designed according to the above conditions. Briefly, the culture in the late-exponential phase was centrifuged at 12000 r/m for 10 min, and the harvested cell pellets were resuspended in fresh MRS broth acidized with 4 mol/L hydrochloric acid to pH 4.5 for acid stress adaption (M20-ASA) or MRS broth containing 0.05% oxgall (BD Difco, Sparks, MD, USA) for bile stress adaption (M20-BSA) at 1 × 10^9^ CFU/mL. They were incubated anaerobically at 37 °C for 1 h. Acid-acclimatized and bile-acclimatized kefir (M20) were tested separately in a controlled trial with unstressed kefir (M20) in MRS broth. The cells with or without the stress adaption treatments were harvested by centrifugation and then resuspended in the same volume of sterile water for the subsequent acid tolerance, bile tolerance, adhesion assay, bile salt hydrolase activity, cellular fatty acid composition and animal experiments.

### 2.3. Assay for Acid Tolerance

The culture was suspended in MRS broth, the pH of which was adjusted to 2.5 with 4 mol/L hydrochloric acid at a concentration of 1 × 10^8^ CFU/mL and cultured at 37 °C under anaerobic conditions for 2 h. The sample (1 mL) was taken out, serially diluted in sterile water and plated onto MRS agar (Oxoid, UK). The plates were incubated for 72 h at 37 °C under anaerobic conditions for colony enumeration. Survival rate of the cells under the acidic environment was calculated using the equation listed below [20].
(1)$$Survival\;rate\;(\%)=\frac{\log\;N_{t}}{\log\;N_{0}}\times 10$$
where N_0_ and N_t_ were the viable counts in the media before and after the incubation (CFU/mL).

### 2.4. Assay for Bile Tolerance

Bile tolerance ability of the cultures was determined by a previous method with minor modifications [21]. Briefly, the cultures were inoculated with one-hundred-thousandth the volume into one-half MRS broth supplemented with and without 0.3% (*w*/*v*) oxgall (BD Difco) that was buffered with 0.1 mol/L sodium phosphate with a final pH of 7.3, and then they were incubated at 37 °C for 12 h. The samples (1 mL) were taken, serially diluted in sterile water and plated onto MRS agar. The plates were incubated for 72 h at 37 °C under anaerobic conditions for colony counting. The bile tolerance of the cultures was calculated by the equation listed below.
(2)$$Bile\;tolerance\;(\%)=\frac{\log_{2}\;\frac{N_{bile}}{N_{0}}}{\log_{2}\;\frac{N_{con}}{N_{0}}}\times 100$$
where N_0_ represents the viable counts before the incubation in the media (CFU/mL) and N_bile_ and N_con_ represent the viable counts after the incubation in the media with and without 0.3% (*w*/*v*) oxgall, respectively.

### 2.5. Assay for Adhesion Ability

Adhesion ability of the cultures was evaluated by a previous method with minor modification [22]. Briefly, Caco-2 cells were cultured in a monolayer on coverslips placed in flat-bottom six-well culture plates that contained 5 mL of Dulbecco’s Modified Eagle Medium (DMEM) supplemented with 20% (*v*/*v*) fetal calf serum from Sigma-Aldrich (St. Louis, MO, USA) at 37 °C in a carbon dioxide incubator. Cells of the cultures were diluted to ~1 × 10^8^ CFU/mL with DMEM, and an aliquot (1 mL) of the diluent was inoculated into the Caco-2 cell monolayer. Following incubation for 1 h at 37 °C, the coverslips were washed three times with phosphate-buffered saline (PBS) to remove unbound bacterial cells [22]. The cells were subsequently fixed with methanol (2 mL) and incubated for 10 min at 37 °C. After evaporating the methanol, the cells were stained with a 1/20 Giemsa stain solution (Sigma-Aldrich) followed by microscopical examination with oil immersion objective. The number of bacteria attached to 100 Caco-2 cells was counted in 20 random microscopic fields.

### 2.6. Assay for Bile Salt Hydrolase Activity

Cell extracts of the cultures were prepared by ultrasonication according to the procedure described by [23]. The cell extracts were used to determine both protein concentrations and bile salt deconjugation ability for calculating specific activity of BSH. The protein concentration was analyzed with a Bradford kit from Bio-Rad (Hercules, CA, USA). Bile salt hydrolase activity was evaluated based on substrate (conjugated bile salts) disappearance rates from a reaction mixture [19]. Briefly, 50 μL of a human bile salt mixture containing 6 conjugated bile acids with a total concentration of 200 mmol/L and 50 μL of an appropriately diluted sample solution were added into 900 μL of a 0.1 mol/L sodium phosphate buffer (pH 6.0). The reactions were carried out at 37 °C. The sample (0.1 mL) was taken after incubation for 10 and 20 min, respectively, and mixed immediately with 50 μL of 15% (*w*/*v*) trichloroacetic acid to terminate the enzymatic reaction. Subsequently, the sample was centrifuged and the supernatant was taken for measuring concentration of the conjugated bile acids by HPLC [24]. One unit of bile salt hydrolase activity was defined as the amount of enzyme which hydrolyzed 1 μmol of conjugated bile salts in the reaction mixture per minute.

Five nanograms of total RNA were isolated from bacterial cultures exposed to bile (0.5% *w/v* Oxgall, sterile PBS, 37 °C, 30 min) and reverse transcribed using qScript cDNA SuperMix (Quanta BioSciences, Gaithersburg, MD, USA). Primer sets were used to generate the standard curve. The genomic DNA of AMC Amplification of 010 was conducted using primer sets BSHqPCR-F (TTGGCGCTGACGACTTGC) and BSHqPCR-R (AATCTTGACGCCTTGACC), which were then employed in RT-qPCR experiments. The qPCR master mix comprised 10 μL of Power SYBR^®^ Green 2x PCR mastermix (Applied Biosystems, Foster City, CA, USA). The master mix comprised 10 μL of Power SYBR^®^ Green 2x PCR master mix (Applied Biosystems, Foster City, CA, USA), 2 μL of primers (BSHqPCR-F, BSHqPCR-R) (1 μM), 1 μL of PCR grade water and 5 μL of template cDNA (1 ng/μL). A total of 40 reaction cycles were conducted, comprising a melting step at 95 °C for 15 s, an annealing/extension step at 65 °C for 45 s and a denaturation step at 95 °C for 10 min prior to SYBR detection. Samples and standards were prepared in triplicate.

### 2.7. Assay for Membrane Fatty Acid Composition

The cultures (1 mL) were mixed with 4 mL of 5% (*w*/*v*) sodium hydroxide solution in 50% aqueous methanol. The mixture was heated at 100 °C for 30 min to saponify the cellular lipids and the released free fatty acids were methylated, followed by extraction with n-hexane according to the procedure reported previously [25]. The formed fatty acid methyl esters were analyzed by GC-MS on a 7890A gas chromatograph fitted with a 5975C mass spectrometer (Agilent Technologies, Palo Alto, CA, USA) and a strong polarity capillary column (DB-WAX, 30 m × 0.25 mm, 0.25 μm of film thickness; Agilent Technologies) with the analytic conditions described previously [26].

### 2.8. Animals and Grouping

Six-week-old male Syrian hamsters were obtained from Beijing Vital River Laboratory Animal Technology Co. (Beijing, China) with permission no. SCXK [Beijing] 2022-0008. The hamsters were housed individually in plastic cages with a 12 h light–dark cycle. Environmental temperature and humidity were maintained at 23 ± 1 °C and 60 ± 5%, respectively. The animals were randomly divided into four groups (*n* = 8 each). All animals were fed a cholesterol-rich diet, which was prepared by adding 0.4% cholesterol into the AIN-93M diet during the 28 d experimental period. Group I (hypercholesterolemic control, HC) was intragastrically given 1 mL of distilled water once daily, while groups II (M20), III (M20-ASA) and IV (M20-BSA) were intragastrically administered 1 mL of non-adapted cells, acid stress-adapted cells and bile stress-adapted cells of L. kefir M20 at 1 × 10^9^ CFU once daily, respectively. All animals were allowed free access to food and water during the 28 d experimental period. The above animal experiments were approved by the China Agricultural University Laboratory Animal Welfare and Ethics Review Committee (Approval Number: AW11013203-1-3).

### 2.9. Assay for Serum Lipids

After hamsters were deprived of feed for 12 h, whole blood was collected from the retro-orbital plexus and serum was separated by centrifugation at 3 kg for 10 min [27]. The obtained serum was analyzed for total and high-density lipoprotein (HDL) cholesterol by an enzymatic colorimetric method using an automated biochemical analyzer (Synchron LX20, Beckman Coulter, Fullerton, CA, USA) with commercially available kits from Biosino Biotechnology and Science Co. (Beijing, China). Non-HDL cholesterol was calculated by subtracting HDL cholesterol from total cholesterol.

### 2.10. Assay for Hepatic Lipids

Following this, the hamsters were euthanized, and their livers were removed, rinsed with physiological saline, dried with filter paper and frozen at 86 °C until analysis. For hepatic lipid analysis, the frozen liver samples (1 g) were homogenized in 5 mL of cold PBS, and the homogenates (1 mL) were taken and extracted three times with 10 mL of chloroform–methanol (2:1, *v*/*v*). The extracts were analyzed for free, and total cholesterol was determined by gas chromatography [28]. Esterified cholesterol was calculated by subtracting free cholesterol from total cholesterol.

### 2.11. Assay for Fecal Bile Acids

Feces were collected continuously during the last 3 d of the feeding period, lyophilized and stored at 86 °C until analysis. For total bile acid analysis, the freeze-dried feces samples (100 mg) were extracted three times with 2 mL of ethanol for 30 min at 60 °C. The combined extracts were evaporated under nitrogen flow and the forming residue was dissolved by 2 mL of ethanol [7]. Total bile acid was determined enzymatically using an automated biochemical analyzer (Synchron LX20, Beckman Coulter) with a commercially available kit from Biosino Biotechnology and Science Co. (Beijing, China).

### 2.12. Statistical Analysis

All the data were expressed as the mean ± SD. Statistical analysis was carried out using IBM SPSS 20.0 (IBM Corp., Armonk, NY, USA). The differences between the means were analyzed by one-way analysis of variance (ANOVA) followed by Duncan’s multiple-range test. The differences were considered significant when *p* < 0.05.

## 3. Results

### 3.1. Effects of In Vitro Acid and Bile Stress Adaption Treatments on Probiotic Properties of the Strain

Both acid and bile stress adaptation treatments significantly improved the acid tolerance of L. kefir M20 (*p* < 0.05). Acid stress adaptation (M20-ASA) was more effective than bile stress adaptation (M20-BSA) in enhancing acid tolerance, with acid stress-adapted cells showing 31.6% higher tolerance and bile stress-adapted cells showing 17.6% higher tolerance compared to non-adapted cells (Figure 1A).

The acid stress adaption treatment did not show a significant (*p* > 0.05) effect on the bile tolerance ability of the strain, whereas the bile stress adaption treatment significantly (*p* < 0.05) enhanced the bile tolerance ability of the strain (Figure 1B). The non-adapted strain was unable to grow in the presence of oxgall, and its viable counts significantly (*p* < 0.05) decreased with direct exposure to the oxgall; however, the bile stress-adapted strain grew better in the presence of 0.3% oxgall.

Neither the acid stress adaption treatment nor the bile stress adaption treatment significantly affected (*p* > 0.05) adhesion of the strain to Caco-2 cells (Figure 1C).

### 3.2. Effects of In Vitro Acid and Bile Stress Adaption Treatments on Bile Salt Hydrolase Activity of the Strain

Acid stress adaptation had no significant effect on bile salt hydrolase (BSH) activity (*p* > 0.05). However, bile stress adaptation significantly increased BSH activity, with a 2.1-fold increase in specific activity compared to non-adapted cells (*p* < 0.05) (Figure 2). Meanwhile, BSH expression levels were quantified by real-time quantitative PCR, which demonstrated a significant enhancement following bile acclimatization treatment. Conversely, no significant difference in BSH expression was observed between acid stress-treated and untreated Kefir M20.

### 3.3. Effect of the Acid and Bile Stress Adaption Treatments on Membrane Fatty Acid Composition of the Strain

Neither acid stress nor bile stress adaptation treatments significantly affected the percentages of membrane 14:0, 16:1, 18:0, 18:1 and 18:2 fatty acids (*p* > 0.05) (Figure 3A,C–F). However, both treatments significantly increased membrane 16:0 fatty acid rates and decreased membrane ∆19:0 fatty acid rates (*p* < 0.05). Acid and bile stress-adapted cells showed 1.52- and 1.50-fold higher membrane 16:0 fatty acid rates, respectively, compared to non-adapted cells (Figure 3B). Non-adapted cells had 1.50- and 1.65-fold higher membrane ∆19:0 fatty acid rates compared to acid and bile stress-adapted cells, respectively (Figure 3G).

### 3.4. Effect of the Acid and Bile Stress Adaption Treatments on Cholesterol-Lowering Ability of the Strain

Neither the non-adapted nor acid stress-adapted cells significantly affected serum HDL-, non-HDL, and total cholesterol concentrations in hamsters (*p* > 0.05), although they showed slightly lower serum total and non-HDL cholesterol concentrations compared to the hypercholesterolemic control (HC) group (Figure 4A–C). Bile stress-adapted cells significantly reduced serum non-HDL and total cholesterol concentrations (*p* < 0.05) but had no significant effect on serum HDL cholesterol concentration (*p* > 0.05). The M20-BSA group showed 33.1% and 16.5% lower serum non-HDL and total cholesterol concentrations, respectively, compared to the HC group, as well as 25.2% and 10.8% lower concentrations compared to the M20 group.

### 3.5. Effect of the Acid and Bile Stress Adaption on Liver Cholesterol-Lowering Ability of the Strain

Neither non-adapted nor acid stress-adapted cells significantly affected hepatic free, esterified and total cholesterol content in hamsters (*p* > 0.05), though they showed slightly lower hepatic esterified and total cholesterol content compared to the HC group (Figure 5A–C). Bile stress-adapted cells significantly reduced hepatic esterified and total cholesterol content (*p* < 0.05), with no significant effect on hepatic free cholesterol. The M20-BSA group had 26.5% and 23.0% lower hepatic esterified and total cholesterol content, respectively, compared to the HC group, and 18.8% and 16.2% lower content compared to the M20 group.

### 3.6. Effect of the Acid and Bile Stress Adaption Treatments on Fecal Bile Acid Excretion in the Animals

Both non-adapted and acid stress-adapted cells promoted fecal bile acid excretion in hamsters, with the M20 and M20-ASA groups showing 1.72- and 1.87-fold higher daily excretion levels compared to the HC group, respectively (Figure 6). Bile stress-adapted cells exhibited a greater ability to promote fecal bile acid excretion than both non-adapted and acid stress-adapted cells. The M20-BSA group showed 9.2-, 5.4-, and 5.0-fold higher daily fecal bile acid excretion levels compared to the HC, M20, and M20-ASA groups, respectively.

## 4. Discussion

This study underscores the lipid-lowering effects of probiotics, specifically Lactobacillus kefir M20. Probiotics modulate lipid metabolism via two key metabolites: short-chain fatty acids (SCFAs) and bile acids (BAs). BAs are pivotal signaling molecules that significantly contribute to fat digestion by emulsifying fats into smaller particles, which are then broken down into fatty acids by lipoprotein lipase. Gastrointestinal tolerance is essential for potential probiotic microorganisms, as they must reach sufficient viable counts in the intestinal tract to exert health benefits [29]. Various strategies have been employed to improve the survival of probiotic strains in the gastrointestinal tract, including protectant application, nutrient addition [30], cell microencapsulation [31] and stress treatments [32]. This study proved the effectiveness of in vitro bile stress adaptation treatment for improving gastrointestinal transit tolerance and enhancing in vivo cholesterol-lowering activity of the strain.

To reach the intestinal tract, ingested strains must first pass through the stomach, where a strong acidic environment with a pH below 1.5 provides a challenge to strain survival [3]. Therefore, gastric acid tolerance is a crucial selection criterion for potential probiotic strains [33]. Bile stress adaptation has been reported to enhance the protein expression level of ATP synthase, a critical protein for maintaining intracellular pH homeostasis in lactobacilli [34]. This mechanism significantly improved the acid tolerance ability of the strain through bile stress adaptation [35]. Additionally, the membrane 16:0 fatty acid has been shown to help bacterial cells resist acid damage in L. casei ATCC334 [36]. In this study, bile stress adaptation increased the membrane 16:0 fatty acid percentage, which may have contributed to the enhanced acid tolerance of the strain.

Bile acids are crucial bioactive compounds in human bile, and due to their detergent nature, human bile is microbicidal, limiting bacterial growth in the intestinal tract [29]. The antimicrobial mechanisms of bile acids include membrane damage and macromolecule stability destruction [37]. Non-stress adapted cells of the strain reduced approximately 2 log orders after 12 h of incubation in broth supplemented with 0.3% oxgall. In contrast, bile stress-adapted cells showed better growth in the presence of 0.3% oxgall, indicating that bile stress adaptation plays a vital role in improving bile tolerance. Bile stress adaptation in lactobacilli has been reported to strengthen the bile efflux system by upregulating proteins dedicated to the active removal of bile acid compounds from the cells [38]. In vitro bile stress adaptation treatment has also been reported to reinforce the cell envelope by altering membrane fatty acid composition in lactobacilli [39]. A positive correlation has been demonstrated between bile tolerance and the relative content of membrane 16:0 and 18:1 fatty acids in lactobacilli [40]. This study observed that bile stress adaptation significantly increased the membrane 16:0 fatty acid percentage, playing a crucial role in preventing membrane damage by bile salts.

Bile stress adaptation also induced bile salt hydrolase (BSH) expression, thereby increasing the specific activity of BSH in L. kefir M20. This was supported by previous observations in L. reuteri CRL1098, where the BSH gene was overexpressed when cells were exposed to glycochenodeoxycholic acid, a type of human bile acid [41]. An in vivo proteomic study showed BSH induction in Bifidobacterium longum NCC2705 when exposed to rabbit large intestine conditions [42]. This study suggested that BSH activity is implicated in bile tolerance. It hypothesized that BSH might be a specific biomarker for bile tolerance in L. kefir M20, supported by a previous study on the cloning and expression of BSH genes in Listeria innocua, which does not encode BSH [43]. Clones expressing BSH showed increased survival in bile and the intestinal tracts of mice compared to the wild-type strain.

This study demonstrated that live cells of L. kefir M20 affect serum cholesterol levels in hamsters by enhancing bile salt deconjugation ability. Enhanced bile acid deconjugation in the small intestine leads to increased fecal bile acid excretion, as deconjugated bile acids have poor water solubility and are less frequently absorbed than their conjugated counterparts [15]. Increased fecal bile acid excretion lowers serum bile acid levels, reducing bile acid transport to the liver. To maintain homeostasis, more new bile acids must be synthesized from cholesterol, significantly decreasing serum cholesterol levels in hamsters [7]. This process may also occur in humans, as hamsters are similar to humans in intestinal bile acid composition [44]. In addition, Lactobacillus induces the down-regulation of key enzymes in bile acid synthesis, such as cholesterol 7α-hydroxylase (CYP7A1) and sterol 12α-hydroxylase (CYP8B1), which ultimately reduces cholesterol levels in the body. In addition, Lactobacillus modulates BSH secretion, which hydrolyses bound bile acids and prevents their absorption, thereby reducing cholesterol levels in the host.

## 5. Conclusions

This study successfully demonstrates that in vitro bile stress adaptation significantly enhances the cholesterol-lowering capabilities of Lactobacillus kefir M20. Adapted strains exhibit improved survival in harsh gastrointestinal environments and show a remarkable ability to reduce serum cholesterol levels in hypercholesterolemic hamsters, achieving reductions in serum total cholesterol and non-HDL cholesterol by 16.5% and 33.1%, respectively. The value of this research lies in its novel approach to enhancing the functional properties of probiotics through targeted stress adaptations, opening new avenues in probiotic research for developing more robust strains capable of surviving gastrointestinal transit and exerting desired therapeutic effects.

The implications of this study are substantial for the functional food industry. The enhanced hypocholesterolemic activity of bile stress-adapted probiotics presents a promising avenue for developing next-generation probiotic products. These products offer a viable, natural alternative to traditional pharmaceutical interventions for managing cholesterol levels, potentially with fewer side effects. Continued research in this area is poised to refine our understanding of the mechanisms through which probiotics interact with host lipid metabolism and expand their application in clinical and nutritional therapies. Ultimately, integrating such probiotic strains into daily dietary practices can become a crucial component of preventative healthcare strategies against cardiovascular diseases. This study underscores the transformative potential of manipulating microbial physiology to enhance probiotic functions. It sets the stage for innovative approaches in the field of functional foods and therapeutic probiotics, with the potential to significantly impact public health and nutrition.

## Figures and Tables

**Figure 1 foods-13-03380-f001:**
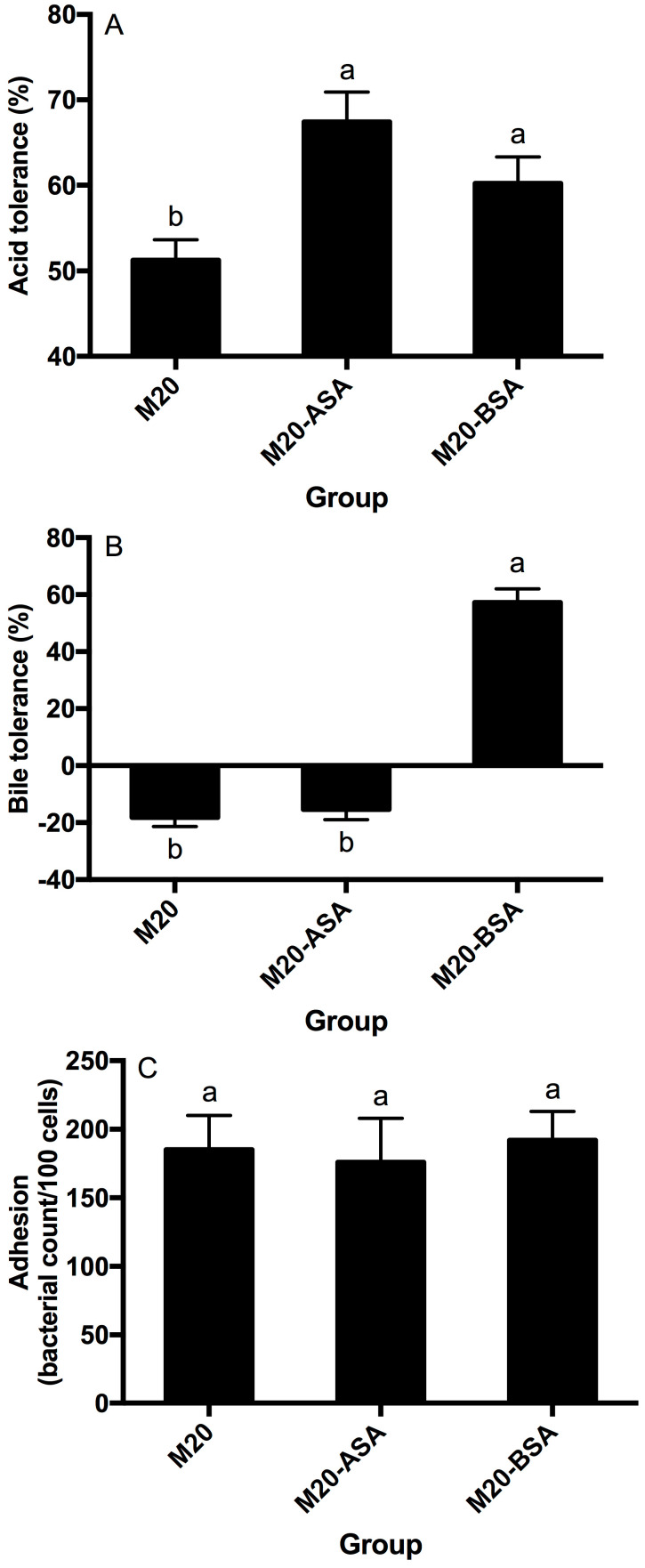
Effects of the acid and bile stress adaption treatments on acid tolerance (**A**), bile tolerance (**B**) and adhesion ability (**C**) of the strain. Data are expressed as the mean ± SD (*n* = 3). Means not sharing a common letter are significantly (*p* < 0.05) different from each other.

**Figure 2 foods-13-03380-f002:**
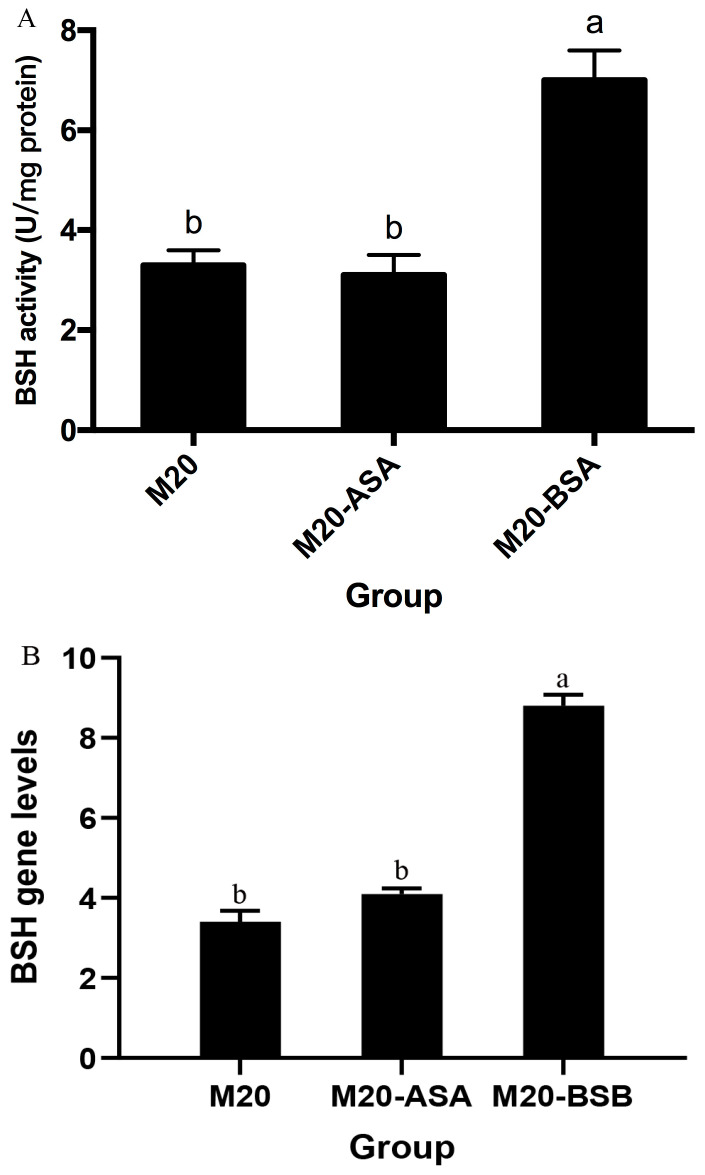
The effects of acid and bile stress acclimation treatments on bile salt hydrolase-specific activity (**A**) and BSH gene expression (**B**) of the strain are presented herewith. The data are presented as the mean ± standard deviation (SD) (*n* = 3). Significant differences were observed between means that did not share the same letter (*p* < 0.05).

**Figure 3 foods-13-03380-f003:**
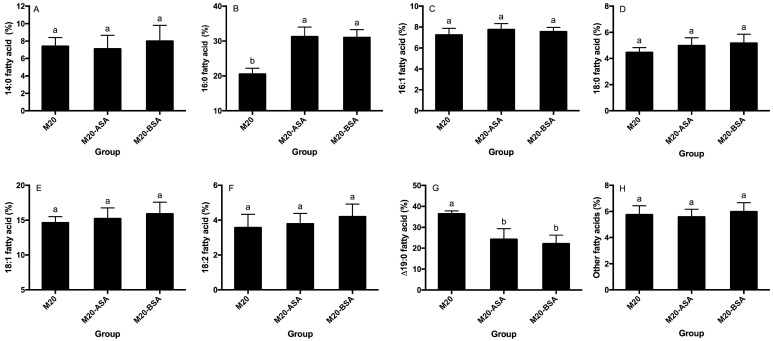
Effects of the acid and bile stress adaption treatments on compositions of membrane 14:0 (**A**), 16:0 (**B**), 16:1 (**C**), 18:0 (**D**), 18:1 (**E**), 18:2 (**F**), Δ19:0 (**G**) and other fatty acids (**H**) of the strain. Fatty acid methyl esters are designated by the number of carbon atoms to the left of the colon and the number of double bonds to the right. Δ, cyclopropane ring. Data are expressed as the mean ± SD (*n* = 3). Means not sharing a common letter are significantly (*p* < 0.05) different from each other.

**Figure 4 foods-13-03380-f004:**
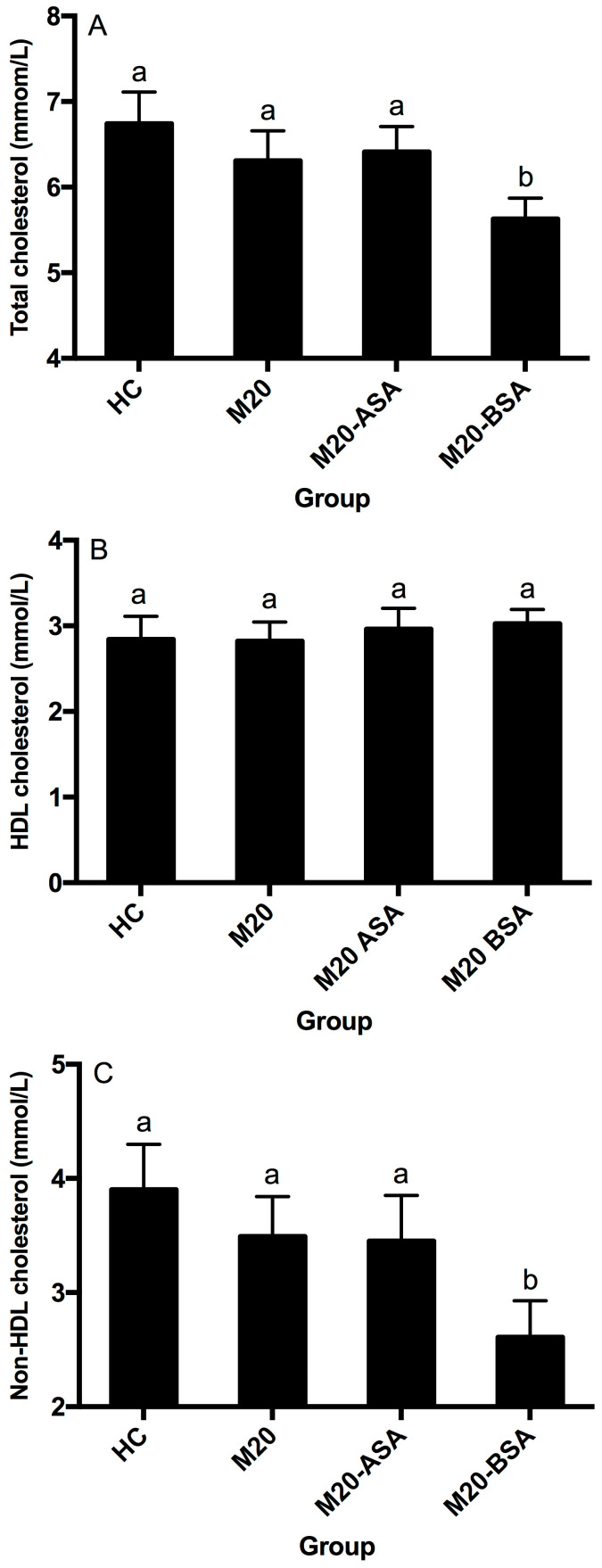
Serum total cholesterol (**A**), HDL cholesterol (**B**) and non-HDL cholesterol (**C**) levels in different groups of hamsters. Abbreviation: HDL, high-density lipoprotein. Data are expressed as the mean ± SD (*n* = 8). Means not sharing a common letter are significantly (*p* < 0.05) different from each other.

**Figure 5 foods-13-03380-f005:**
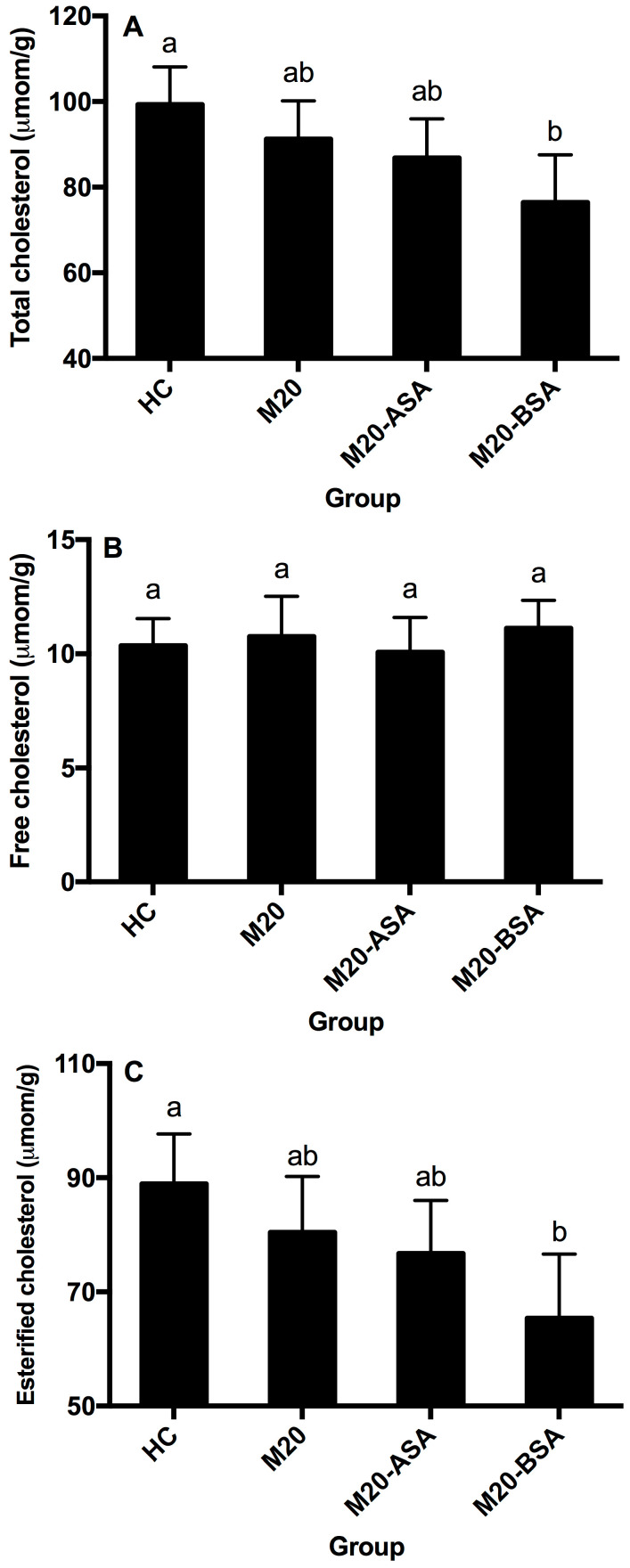
Hepatic total cholesterol (**A**), free cholesterol (**B**), and esterified cholesterol (**C**) levels in different groups of hamsters. Data are expressed as the mean ± SD (*n* = 8). Means not sharing a common letter are significantly (*p* < 0.05) different from each other.

**Figure 6 foods-13-03380-f006:**
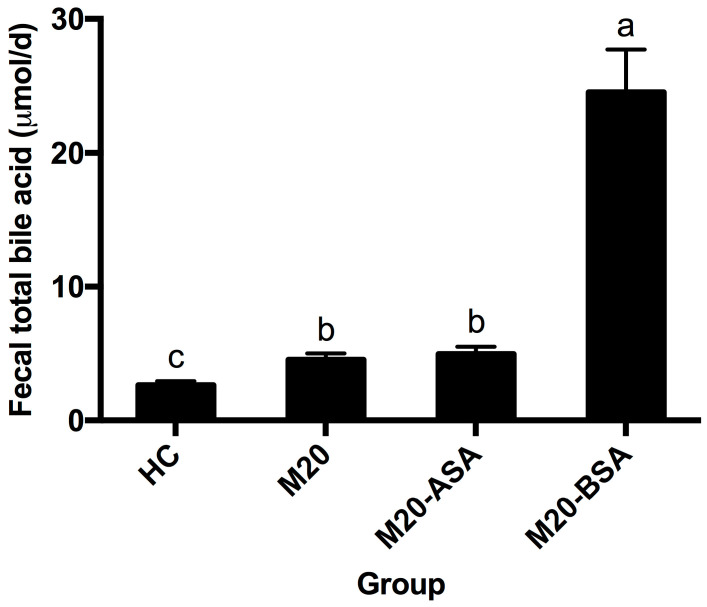
Daily fecal bile acid excretion levels in different groups of hamsters. Data are expressed as the mean ± SD (*n* = 8). Means not sharing a common letter are significantly (*p* < 0.05) different from each other.

## Data Availability

The original contributions presented in the study are included in the article, further inquiries can be directed to the corresponding author.

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
