# Peer review of "Lactobacillus Kefir M20 Adaptation to Bile Salts: A Novel Pathway for Cholesterol Reduction"

_foods, 2024, doi:10.3390/foods13213380_

Round 1

Reviewer 1 Report

Comments and Suggestions for Authors 1-Lactobacillus kefir 1; acid stress adaptation 2; bile stress adaptation 3; cholesterol-lowering 4; Hamster 5--number? 2-The introduction section needs to include more information on cholesterol-lowering. 3-Please add the significance of the sentence. The strain was trans-63 ferred three times in MRS broth using the aforementioned incubation conditions. 3-Is this strain isolated aerobically or anaerobically? 4-Please note that we conducted the pH of the A control test in MRS broth without administering any stress treatment (M20). line-L. kefir M20 151 at 1 × 109 CFU once daily, respectively, need to correct

5-List the authority that provided approval and the corresponding ethical approval code. Comments on the Quality of English Language

minor  correction required

Author Response

Comment 1:1-Lactobacillus kefir 1; acid stress adaptation 2; bile stress adaptation 3; cholesterol-lowering 4; Hamster 5--number? 

Thank you for pointing this out, we acknowledge your comment and have therefore made changes to the keywords section in lines 20-21 to remove the incorrect number.

Comment 2:2-The introduction section needs to include more information on cholesterol-lowering.

Thank you for pointing this out,We have revised the Results section in the Introduction of the manuscript (lines 12-16) to add information on cholesterol lowering by Kaempfer M20 after stress-adaptive treatment in mouse experiments.

Comment 3:3-Please add the significance of the sentence. The strain was trans-63 ferred three times in MRS broth using the aforementioned incubation conditions. 3-Is this strain isolated aerobically or anaerobically?

Thanks to your valuable suggestion, I have reworded this sentence in lines 64-66 of the text to express the culture conditions of Kefir M20.

Comment 4:4-Please note that we conducted the pH of the A control test in MRS broth without administering any stress treatment (M20). line-L. kefir M20 151 at 1 × 109 CFU once daily, respectively.

Thank you for bringing this to our attention. Following your suggestion, we have amended the original sentences in lines 77-78 and line 167 of the article to make them clearer.

Comment 5:Need to correct 5-List the authority that provided approval and the corresponding ethical approval code.

We are grateful for your observation and have augmented the article's rigor by including the approving organizations and corresponding ethical approval codes in lines 167-170.

Reviewer 2 Report

Comments and Suggestions for Authors

General comments

This study is focused on evaluating the impact of in vitro adaptations to acid and bile stress on the cholesterol-lowering activity of Lactobacillus kefir M20. The in vitro and in vivo results are interesting and have potential implications for the successful design of functional supplements capable of lowering serum cholesterol. However, some specific issues should be carefully reviewed.

 Specific comments

Please note that Lactobacillus kefir is now called Lentilactobacillus kefiri in accordance with Zheng et al., (2020, A taxonomic note on the genus Lactobacillus: Description of 23 novel genera, emended description of the genus Lactobacillus Beijerink 1901, and union of Lactobacillaceae and Leuconostocaceae.https://doi.org/10.1099/ijsem.0.00410)

What are the studies that support the probiotic character of Lactobacillus kefir  M20? They should be correctly cited in the introduction and in the M&M to justify it being considered probiotic. The papers cited in the introduction do not seem to justify the strain being considered probiotic but potentially probiotic.

Throughout the text check the correct use of italics.

L 43-45: check reference as it does not coincide with the quotation.

L 52: should say that many bacteria isolated from kefir have demonstrated probiotic properties.

The results in Figures 1b and 1c are not described in the results.

To make the graphs self-reading, please clarify the terms M20-ASA and M20-BSA in the figure legends.

297-298: On the basis of what results do you suggest that the BSH enzyme was overexpressed? Your assay results indicate increased BSH activity which may be due to different mechanisms and not necessarily to overexpression. To evaluate gene overexpression you should perform Real Time PCR or transcriptomic assays while for protein expression proteomic assays are required.

L 315: Lactobacillus does not induces activation of the Farnesoid X receptor, but bile acids.

L318: Lactobacillus spp. does not secrete BSH, since in most BSH-positive strains of Lactobacillus and related genera (including those cited in this work), the enzymes are intracellular.

Author Response

Comment 1:L 43-45: check reference as it does not coincide with the quotation.

Thank you for pointing out the problem, the relevant references have been included in sections 14 and 15 to ensure that the content of the article is aligned with the subject matter.

Comment 2:should say that many bacteria isolated from kefir have demonstrated probiotic properties.The results in Figures 1b and 1c are not described in the results.To make the graphs self-reading, please clarify the terms M20-ASA and M20-BSA in the figure legends.

We are grateful for your guidance, which has enabled us to enhance the rigour of the article by adding the notes M20-ASA and M20-BSA to lines 202 and 203.

Comment 3:297-298: On the basis of what results do you suggest that the BSH enzyme was overexpressed? Your assay results indicate increased BSH activity which may be due to different mechanisms and not necessarily to overexpression. To evaluate gene overexpression you should perform Real Time PCR or transcriptomic assays while for protein expression proteomic assays are required.

In accordance with your recommendation, we have incorporated the qPCR assay into lines 134-146 of the article. Concurrently, we have included the real-time fluorescence PCR results and data analysis in lines 222-228 of the results section of the article, thereby enhancing the article's overall completeness.

Comment 4: L 315: Lactobacillus does not induces activation of the Farnesoid X receptor, but bile acids. L318: Lactobacillus spp. does not secrete BSH, since in most BSH-positive strains of Lactobacillus and related genera (including those cited in this work), the enzymes are intracellular.

We are grateful for your guidance, which enabled us to re-edit lines 344-348 of the manuscript. The cholesterol-lowering mechanism of lactobacilli was reorganised to ensure the scientific validity of the article.